



# Surface transport of DOC acts as a trophic link among Mediterranean sub-basins

Chiara Santinelli[1,3], Roberto Iacono[2], Ernesto Napolitano[2], and Maurizio Ribera d'Alcalá[3]

[1]CNR, Istituto di Biofisica, Pisa, Italy
[2]ENEA, C. R. Casaccia, Rome, Italy
[3]Stazione Zoologica Anton Dohrn, Napoli, Italy

**Correspondence:** Santinelli (chiara.santinelli@pi.ibf.cnr.it)

**Abstract.** Characterizing carbon cycling and redistribution in the ocean is an important issue for Mankind, because it may affect key ecosystem services, e.g., support to climate system and food provision. In this paper, using an integrated approach, we explore the impact of the surface circulation on carbon dynamics in the Western Mediterranean Sea, where strong inter-basin differences in primary production do exist. Detailed information on the surface circulation, derived from high-resolution model simulations, is combined with the analysis of accurate, repeated dissolved organic carbon (DOC) data. Our work indicates that the advection of the Atlantic Water acts as a trophic link between the Algerian Basin and the Tyrrhenian Sea, determining a flux of $8.8 - 37.9 \cdot 10^{12} \ g \ DOC \cdot yr^{-1}$ into the basin. Thus, surface transport of DOC can redistribute chemical energy among regions with different trophic regimes. We hypothesize that this overlooked mechanism plays an important role also in the global ocean.

## 1 Introduction

Dissolved Organic Carbon (DOC) in the oceans is one of the largest reservoirs of reduced carbon on Earth and supports heterotrophic prokaryotes respiration. Various fractions have been distinguished in its pool depending on their lifetime, going from hours-days (Labile DOC, LDOC), to months-years (Semi-Labile DOC, SLDOC), and up to thousands years (16,000 years, Refractory DOC, RDOC; 40,000 years, Ultra-Refractory DOC, URDOC) (Hansell, 2013). From an ecological and biogeochemical perspective, the SLDOC plays the most intriguing role, because it escapes rapid microbial removal, accumulates in the water, and can therefore be transported over long distances before being recycled. Due to the interest in quantifying the carbon sinks in the ocean, several studies focused on the vertical transport of DOC by winter mixing and deep water formation in both the oceans (Carlson et al., 1994, 2010) and the Mediterranean Sea (Avril, 2002; Santinelli et al., 2013) and showed that the exported DOC can support respiration in the dark ocean (Arístegui et al., 2003; Santana-Falcón et al., 2017). Less attention has been paid to the role of horizontal transport in DOC redistribution among basins, even though it has long been recognized that the SLDOC, accumulated in the surface layer, can be transported for long distances by currents, such as the Gulf Stream (Hansell and Carlson, 1998). The surface layer receives external inputs from rivers, atmosphere and, in case of marginal seas, from the neighboring oceans. It is the place where most of the DOC is produced, transformed and removed. Therein, DOC can accumulate and become available to be exported and eventually sequestered to depth. DOC fluxes within this layer are there-





fore crucial, but their estimation is very complex, because of the short time scales of the numerous biogeochemical processes affecting DOC concentration. Hansell et al. (1997) observed that approximately $20\%$ of net community production is advected horizontally as Total Organic Carbon (TOC) in the equatorial Pacific Ocean. Lomas et al. (2010) highlighted the importance of the advection of Dissolved Organic Phosphorus (DOP) in the P-budget of the Sargasso Sea, estimating a transport rate into the

Bermuda Time Series Station (BATS) region as high as $43\ mmol\ DOP\ m^2 \cdot yr^{-1}$. More recently, Wu et al. (2015) showed that the Kuroshio intrusion plays a dominant role in determining the TOC inventory and distribution in the central Northern South China Sea. Santana-Falcón et al. (2016) estimated that the total organic carbon exported yearly from the Cape Ghir upwelling filament represents at least $29\%$ of the primary production in this area. DOC distribution in the surface layer is also affected by eddies. DOC accumulation is usually observed in the core of anticyclones (Moutin and Prieur, 2012; Arístegui et al., 2003;

Mathis et al., 2007) and it is associated with elevated microbial respiration and heterotrophic production (Baltar et al., 2010). In contrast, cyclonic eddies are characterized by low DOC concentrations, due to the doming of intermediate waters (Santinelli et al., 2008). Disentangling the role that the different dynamic structures play in shaping the DOC distribution is challenging, but of crucial importance, since it allows a better quantification of the biogeochemical processes influencing DOC distribution. Herein, we address this issue for the Western Mediterranean Sea (WMED), using an integrated approach. Detailed informa-

tion on the Tyrrhenian Sea (TYS) circulation derived from high-resolution numerical models is used to interpret the spatial distributions of DOC measured in the southern TYS during different seasons (2006-2011).

## 2 Data and methods

### 2.1 DOC measurements

During 8 cruises in the southern TYS (November 2006, April and June 2007, January 2009, February, August and October

2010, November 2011), seawater samples were collected along a 6-station section (4-station in April 2007 and 3-station in February 2010, not shown) (Fig. 1 and 2). At all stations, pressure, conductivity and temperature were measured by a SBE 911plus CTD, equipped with a rosette sampler.

Samples were immediately filtered through $0.2\ \mu m$ nylon membrane; DOC measurements were carried out with a Shimadzu TOC-Vcsn. Measurement reliability was assessed twice a day by comparison with DOC Consensus Reference Waters (CRM)

(Hansell, 2005) (batch # 7, 8, 9 and 10, measured concentrations: $42.1 \pm 1.3\ \mu M$, standard error $0.10\ \mu M$, $n = 166$). The analytical precision was better than $1.5\%$. For further analytical details see Santinelli et al. (2015). Average concentrations of DOC, weighted over depth, were calculated in different layers ($0 - 50\ m$ layer in Table 1) taking into consideration the individual concentrations in a given profile weighted by the depth interval, using the trapezoid rule, as follows

$$DOC_{Z_0-Z_n} = \frac{\sum_1^n \frac{(DOC_{Z_i}+DOC_{Z_{i-1}})}{2}(Z_i - Z_{i-1})}{Z_n - Z_0}$$

Where $DOC_{Z_i}$ is the concentration of DOC at the depth $Z_i$. We collected 3-4 samples in the layer $0 - 50\ m$, depending on the cruise. The DOC stocks were calculated multiplying the depth averaged concentration by the thickness of the layer.



Uncertainties were estimated taking into consideration the error on DOC measurement (that ranged between 0 and 1.6 $\mu M$) and applying the propagation of errors rule. The uncertainties ranged between 0.8 and 1.7%.

### 2.1.1 Model and altimeter data

The numerical results used in this study are current, temperature and salinity fields produced by a recent high-resolution
model of the TYS circulation (TYREM operational model; $1/48° \times 1/48°$ horizontal resolution, 40 sigma levels), based on the Princeton Ocean Model (POM) (Mellor, 1998) which is described in detail in Napolitano et al. (2014, 2016). The skill of the model has been evaluated in the cited references, through model-data comparison. Maps of the absolute dynamic topography (ADT), with the corresponding geostrophic currents, are obtained from AVISO data (http://www.aviso.oceanobs.com) (Fig. 1).

## 3  Results and Discussion

Before discussing results, we briefly recall some basic elements of the surface circulation in the WMED (Astraldi et al., 1999; Millot and Taupier-Letage, 2005). This circulation is dominated by the inflow of the Atlantic Water (AW) from the Gibraltar Strait, with a salinity that is much lower than the average salinity of the WMED. The AW borders the North-African coasts, crosses the Sardinia Channel, and then follows different paths, depending on the season (Fig. 1). In winter it bifurcates; a portion flows into the Eastern Mediterranean Sea (EMED) through the Sicily Channel, whereas a robust stream enters the
TYS. This stream, sustained by a large-scale cyclonic wind forcing, circulates cyclonically along the Italian coasts and finally exits the TYS through the Corsica Channel (Fig. 1a). In spring, the global cyclonic wind stress curl weakens, and the flow becomes unstable, forming several anticyclonic gyres along the Italian coasts (Iacono et al., 2013; Napolitano et al., 2014). In summer, the wind stress switches to anticyclonic rotation in the southern TYS, so that a global anticyclonic circulation settles in the area, preventing the AW from entering the basin (Fig. 1b). At the same time, the anticyclonic circulation in the area near
the Corsica Channel isolates the TYS to the North, preventing exchanges with the Liguro-Provencal basin (Ciuffardi et al., 2016). In this period, persistent wide mesoscale anticyclonic structures develop in the southern TYS.

### 3.1  DOC vertical distribution

DOC vertical distribution shows a clear seasonality in the upper 200 $m$ of the water column. Throughout the year, DOC shows the highest concentrations ($50-75$ $\mu M$) in the surface layer (0-100 m), a decrease in correspondence with the pycnocline
and values of $40-45$ $\mu M$ below it. Remarkably, concentrations higher than 65 $\mu M$ are observed in all the seasons, except in winter (Fig. 2). DOC averages, calculated by vertical integration in the upper 50 $m$ (Table 1), show the lowest values in winter ($56.9 \pm 0.6$ $\mu M$ in January 2009 and $58.7 \pm 0.7$ $\mu M$ in February 2010) ($0-50$ $m$ DOC stocks: $2.84 \pm 0.03 - 2.93 \pm 0.04$ $mol \cdot m^{-2}$). All the other periods, excluding November 2006, are characterized by averages between $62.4 \pm 0.9$ and $65.4 \pm 0.5$ $\mu M$ ($0-50$ $m$ DOC stocks: $3.12 \pm 0.05$ to $3.27 \pm 0.03$ $mol \cdot m^{-2}$). November 2006 shows values lower ($59.3 \pm 1.0$ $\mu M$) than
November 2011 ($65.4 \pm 0.5$ $\mu M$). Differences can be observed in the spatial distribution of DOC along the section (Fig. 2 and Standard Deviation (SD) in Table 1). April 2007 is characterized by high DOC average concentrations ($65.2 \pm 1$ $\mu M$),





with values of $\sim 70\ \mu M$ at the central stations, and the highest SD ($6.4\ \mu M$). In June 2007, August and October 2010 and November 2011, the highest values are found at the edge of the section. In winter, DOC is homogeneously distributed, with a SD of $2\ \mu M$ in January 2009 and $0.1\ \mu M$ in February 2010 (only 3 stations sampled), when a deeper mixed layer is also observed (See isopycnals in Fig. 2). These data indicate a $0.33 - 0.42\ mol \cdot m^{-2}$ increase in DOC stocks between winter and

spring (January/February to April) and a DOC redistribution between spring and autumn (April to October), when the stocks are comparable (Table 1). The seasonal difference is 7-11 time larger than uncertainties. In the following, we will focus on key periods, showing features that highlight the link between physical dynamics and biogeochemical processes.

## 3.2  DOC in spring, the role of the AW

April 2007 is characterized by high DOC values, with a marked accumulation in the central stations (stations VTM2 and VTM3,

Fig. 2, 3c, 3d and Table 1). The $\theta - S$ diagram shows DOC values of $54 - 64\ \mu M$ in correspondence with the salinity minimum at the stations VTM5 and VTM ($0 - 65\ m$) (Fig. 3c). The circulation in the area, as simulated by TYREM, can explain the DOC spatial distribution. Surface velocity (Fig. 3a) and salinity (Fig. 3b) fields as well as their vertical distribution (Fig. 3d) highlight the occurrence of a robust AW stream, in the upper $200\ m$ of the offshore stations. The highest DOC concentrations (70-73 μM) occur in the AW core ($0 - 80\ m$ of station VTM2) and in the upper 50 m of station VTM3, that lies at the border between

the AW stream and an anticyclonic re-circulation in the near-shore area. Values of $56 - 64\ \mu M$ are observed in the upper $50\ m$ of station VTM, located at the edge of the stream, where the velocity sharply decreases to zero, and at station VTM5, located outside the core of the AW, where mixing with local water occurs (Fig. 3c and 3d). These data suggest that the AW stream plays a crucial role in shaping DOC distribution. Considering the typical velocities of the Algerian current, it can be estimated that the AW takes about 2 months to go from the Algerian Basin to the eastern TYS. The AW detected in April in the TYS

was therefore in the Algerian Basin in February, when winter phytoplankton blooms usually occur (Mayot et al., 2016), with primary production rates ($113 - 160\ g\ C \cdot m^{-2} y^{-1}$) higher than in the TYS ($90 - 92\ g\ C \cdot m^{-2} y^{-1}$) (Lazzari et al., 2012). The evolution of the surface chlorophyll-a from January to April 2007 (Fig. 4) confirms the occurrence of the bloom in February 2007. Thus, in April 2007 it is likely that the high DOC concentrations mostly result from the transport of the DOC, produced outside the basin, by the AW stream. Since in 2007 the TYS winter-spring dynamics was quite typical (Iacono et al., 2013), we

expect the remote transport to provide a regular DOC feeding mechanism for the TYS in spring. It is worth mentioning that the AW stream is, in some regions, very wide, therefore DOC concentration can be not homogeneous, depending on the dominant biological processes in the various areas of the stream. As an example, the area of the stream hosting a phytoplankton bloom is expected to have higher DOC concentrations than the area where the primary production is low.

## 3.3  DOC in summer, the June 2007 and August 2010 cases

In summer, the AW does not flow into the TYS, which becomes basically isolated (Fig. 1b). Surface DOC concentration remains high between April and October (Fig. 2), with a small change of its average ($62.4 \pm 0.9$ in June 2007 and $64.4 \pm 0.7$ in August 2010) (Table 1). The section crosses a wide anticyclonic circulation, a recurrent feature for the area in summer (Rio et al., 2014), and the highest DOC concentrations are found at the edges of the anticyclone. These observations point



to a role of the anticyclone in shaping the horizontal distribution of DOC, possibly by two different, not mutually exclusive, mechanisms; (1) the anticyclone may determine dynamical regions where DOC can accumulate; (2) the anticyclone, extending to the coastal zone, can bring into the open sea the DOC either of terrestrial origin or produced in-situ during the coastal blooms and post-bloom phases, accumulated in the coastal area.

Mechanism 1 relies on the assumption that DOC is a passive tracer on the temporal scale of months; as such, its concentration can be strongly affected by the geometry of the advecting velocity field. In a quasi-steady two-dimensional flow, regions that are attracting (increase in the tracer concentration) or repelling (decrease in the tracer concentration) can be identified through the value of the lambda parameter ($\lambda$), introduced by Haller and Iacono (2003); $\lambda$ is negative in attracting regions. In August 2010, when the circulation is fairly stable, the section cuts a wide anticyclone, a semi-permanent feature of the summer circulation
in the area (Iacono et al., 2013) (Fig. 5). The anticyclone is in its turn inserted into a surrounding, wider anticyclonic cell that occupies the whole southeastern portion of the TYS, which may be expected to entrain the DOC (of remote origin + locally produced) present in the area. Moving around, the DOC crosses two regions, to the northwest and south-southeast of the section, with strongly negative values of $\lambda$ (green areas in Fig. 5), where the areal concentration can increase. These regions border the inner cell, horizontal mixing may therefore increase DOC concentration on the two sides of this cell consistently
with the observations (Fig. 2). This mechanism occurs everywhere in the oceans, but the impact that the gradients of resources (such as DOC) have on the function of the food web has seldom been analyzed in detail.

### 3.4   DOC in late autumn: the November 2006 and November 2011 case

As previously reported, DOC distribution is markedly different in November 2006 and 2011. Average DOC concentrations $(0-50\,m)$ are $\sim 6\,\mu M$ lower in 2006 than in 2011, and this difference results in $\sim 0.31\,mol\cdot m^{-2}$ reduction in the $0-50\,m$
accumulated stocks (Table 1). This difference is of the same order of magnitude as the annual variability $(0.33-0.43\,mol\cdot m^{-2})$. In November 2011, a typical late-autumn pattern was observed, with a robust AW global cyclonic cell already formed, whereas through most of 2006 the circulation was quite weak in the eastern TYS, resulting in little AW inflow into the TYS in November (Fig. 6). The energy minima in 2006 (from May to November) (Fig. 6a) correspond to an anomalous increase in the average surface salinity, that reaches a maximum value of $\sim 38.35$, higher than the maxima of the following years (Fig. 6b). It is
noteworthy that the difference in salinity between November 2006 and 2011 is comparable to the annual variation. These observations indicate that the freshening of the southern TYS, due to the AW inflow, is strongly reduced in 2006 and this corresponds to a markedly lower inflow of DOC, explaining the lower DOC concentration observed in November 2006 than 2011.

### 3.5   DOC annual cycle

The tight link between the TYS surface circulation and the annual DOC cycle can be summarized as follows:

1. Late winter to spring, DOC shows a marked increase that can be mainly explained by the inflow of AW rich in DOC likely due to biological production resulting from the phytoplankton blooms in the Algerian Basin.





2. Spring to early summer, the DOC, accumulated in the surface layer, is redistributed by the surface circulation within the basin. In this period, the anticyclone in the southern TYS forms and stabilizes.

3. Summer to mid-autumn, DOC concentration in the mixed layer remains high; DOC production and removal are apparently coupled since no change in DOC concentration is observed. DOC concentration is higher at the edge of the anticyclone. This is the time of the year when regenerated production dominates (Ribera d'Alcalà et al., 2009). With the available data-set, we cannot determine if microbial respiration uses the DOC produced in-situ in spring or that imported from the WMED by the AW inflow. Even if DOC concentration does not change, DOC turnover can occur either by (1) advected DOC is consumed and replaced by DOC released within the basin or coming from external sources or (2) removal of DOC released within the basin or coming from external sources and aging of the advected DOC.

4. Late autumn to beginning of winter, the global cyclonic cell settles again, washing out DOC from the southeastern basin determining low DOC surface concentrations and almost no horizontal gradient.

### 3.6 DOC stocks and fluxes in the Tyrrhenian Sea

In order to determine the overall impact of the AW advection in the C cycle of the TYS, DOC stocks and fluxes were estimated using the data and the model outputs. DOC stocks were estimated by multiplying the mean DOC concentration in the surface, intermediate and deep layers by the volume of the three layers (Table 2). The resulting total DOC stock for the whole basin is $204 - 246 \cdot 10^{12}$ $g$ DOC ($7.4 - 9.4 \cdot 10^{12}$ $g$ DOC in the upper $50\,m$). Assuming that the DOC concentration in the deep water ($\sim 41\,\mu M$) represents the less reactive fraction (e.g., RDOC+URDOC), $2.4 - 3.4 \cdot 10^{12}$ $g$ DOC would represent the SLDOC in the surface layer ($0 - 50\,m$). This would be the fraction available on the short temporal scale (months) for microbial respiration and would, therefore, play the most important role from both an ecological and biogeochemical perspective (**?**). This fraction escapes rapid microbial removal, thus becoming a reservoir of energy that may be transported in areas far from its production site by surface circulation as observed in upwelling regions of the oceans (Santana-Falcón et al., 2016) and, if it reaches specific locations (e.g. the Gulf of Lions) it can be exported to depth by winter convection (Santinelli, 2015; Santinelli et al., 2010).

In the TYS, DOC can be produced in-situ and/or supplied by rivers, atmosphere and AW advection. Considering that the primary production, estimated for the TYS, ranges between 90 and 92 $g\,C \cdot m^{-2}yr^{-1}$ (Lazzari et al., 2012) and assuming that $13 - 35\%$ is released as DOC by phytoplankton and that additional $25\%$ can be released by zooplankton (Carlson and Hansell, 2015), this would account for a DOC in-situ production of $11.7 - 55.2$ $g\,C \cdot m^{-2}yr^{-1}$, that is $2.6 - 12.3 \cdot 10^{12}$ $g\,C \cdot yr^{-1}$ on the basin-scale. This amount of DOC is not enough to satisfy the prokaryotic carbon demand estimated for the euphotic layer of the TYS ($13 - 21 \cdot 10^{12}$ $g\,C \cdot yr^{-1}$) (La Ferla et al., 2010; Santinelli et al., 2012). Due to the semi-enclosed nature of the basin, terrestrial inputs should be important in the TYS; the total water discharge is $19.4\,km^3 \cdot yr^{-1}$ and accounts for a DOC input of $0.06 \cdot 10^{12}$ $g\,C\,yr^{-1}$ (Santinelli, 2015). No information about atmospheric input is at present available for this basin, but, extrapolating the little information available for the Mediterranean Sea, it may account for $0.05 - 1.45 \cdot 10^{12}$ $g\,C\,yr^{-1}$ (Santinelli, 2015). The AW fluxes, estimated by the model, ranged between $0.46\,Sv$ (November 2006) and $1.78\,Sv$ (February 2010) (Table 3). Taking into consideration the average concentration of DOC in the upper 200 m in the same period, we can



estimate that AW advection can supply the Tyrrhenian Sea with $8.8 - 37.9 \ 10^{12} \ g \ DOC \ yr^{-1}$. These data suggest that AW advection may supply the TYS with an amount of DOC larger than that produced in-situ, strongly supporting the importance of this process in feeding the basin. A similar observation has been reported in the oceans where upwelling region can feed oligothrophic areas through the transport of organic carbon by filaments (Santana-Falcón et al., 2016). It is noteworthy that the

AW flux in November 2006 is similar to summer values, determining a DOC input 2-fold lower in November 2006 than 2011. This observation highlights that changes in the circulation pattern can have a strong impact on marine ecosystem supplying a different amount of DOC.

## 4    Conclusions

Our analysis demonstrates that surface advection of AW plays a crucial role in shaping DOC distribution in the TYS and in

modulating the carbon exchange among sub-basins. Advected AW can supply the TYS with an amount of DOC larger than that produced in-situ by primary production making the Algerian Basin a resource feeder for the TYS. This supply may have a large interannual variability, which is of the same order of magnitude as the seasonal variations, significantly affecting the annual carbon cycle of the basin. This process also raises intriguing questions about the impact of the AW transport on the extreme oligotrophy of the EMED. While the small size of the Mediterranean Sea may favor the above process, it is likely

that also in other regions of the oceans, surface advection may set up, via horizontal transfer, a sort of compensation among regions with different trophic regimes. All the above clearly indicates that the quantification of horizontal DOC fluxes should be considered among the outputs of numerical models simulating the global carbon cycle.

*Author contributions.* CS generated DOC data, RI and EN run the model simulations, CS, RI, EN and MRdA designed the study and wrote the manuscript

*Acknowledgements.* Physical data were kindly provided by the "Dipartimento di Scienze per l'Ambiente, Università Parthenope" (E. Zambianchi), "Dipartimento di Scienze del Mare, Università Politecnica delle Marche" (A. Russo), "CNR-IAMC" (M. Sprovieri), "CNR-ISAC" (R. Santoleri), "CNR-ISMAR", (M. Borghini). This research was supported by the Italian Flagship projects VECTOR and RITMARE funded by the Ministry of Research and University and the PERSEUS project, VII FP, OCEAN.2011-3, (No. 287600).



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



**Figure 1.** Study area and sampling stations superimposed on the average maps of ADT (colors) and geostrophic velocity (black arrows), both derived from AVISO data. The white arrows indicate the main paths of the Atlantic Water





**Figure 2.** DOC [$\mu M$] vertical distribution in the upper 200 $m$ along the section reported in Figure 1 and in the map of the area. The black lines are isopycnals





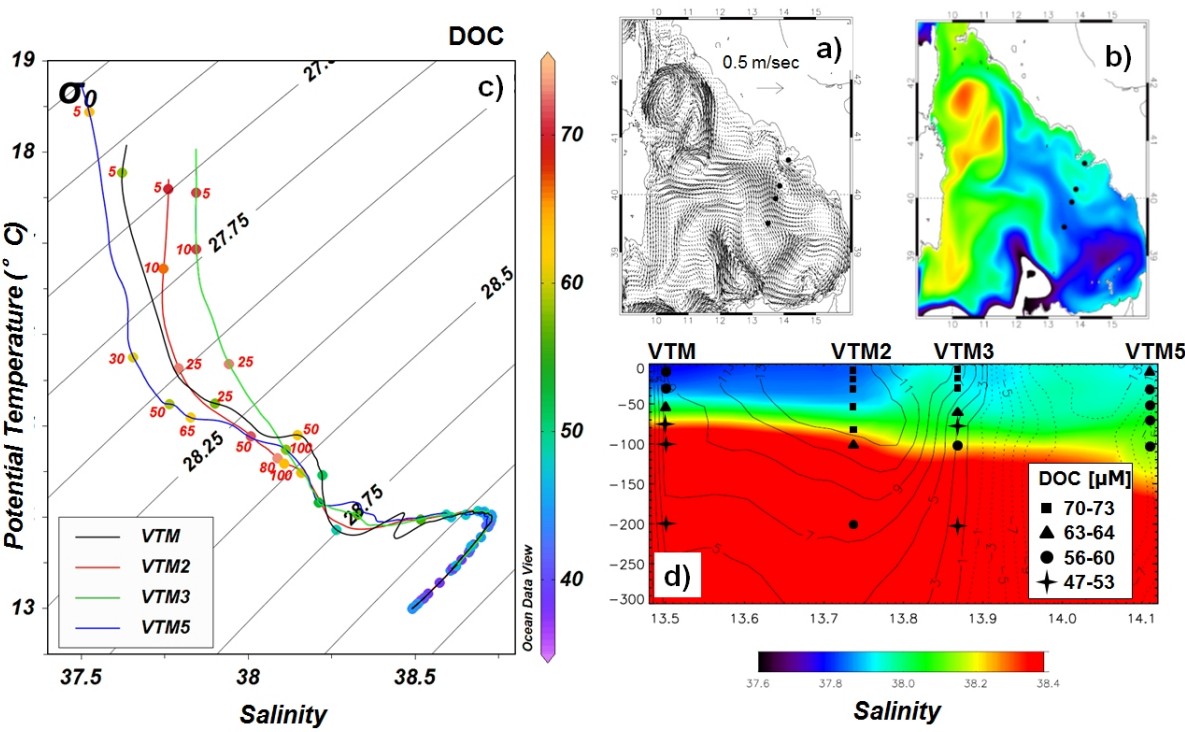

**Figure 3.** Surface velocity (a) and salinity (b) fields averaged over the upper $50\ m$ (TYREM model); $\theta - S$ graph with superimposed DOC concentration (colors) and the sampling depths (red numbers) (c); vertical distribution of salinity and velocity component normal to the section (d) (TYREM model) (solid lines refer to velocity toward northwest), symbols refer to the DOC concentrations. All the data refer to April 2007



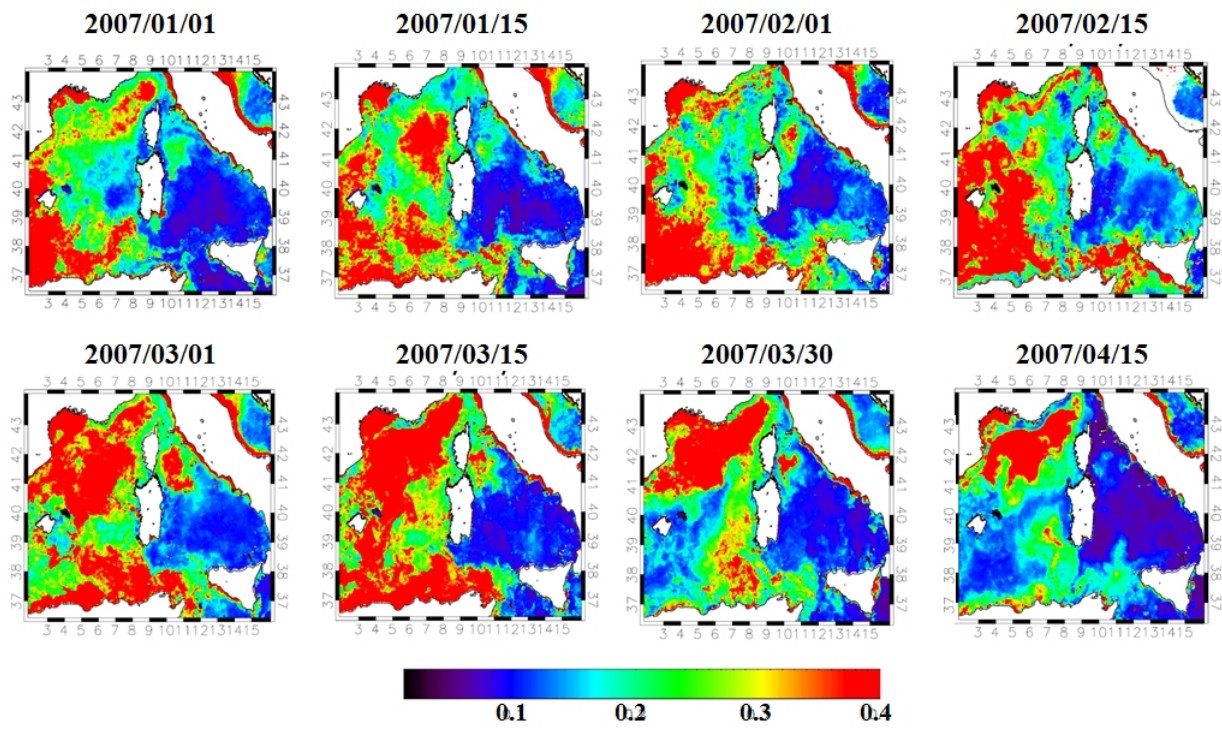

**Figure 4.** Mediterranean Sea Surface Chlorophyll Concentration from Multi Satellite observations reprocessed by ISAC–CNR, Rome, Italy.

The data set is available at Copernicus Web Site (http://marine.copernicus.eu)





**Figure 5.** Circulation in the TYS in August 2010 (TYREM hindcast), colors refer to the values of the transport parameter (λ

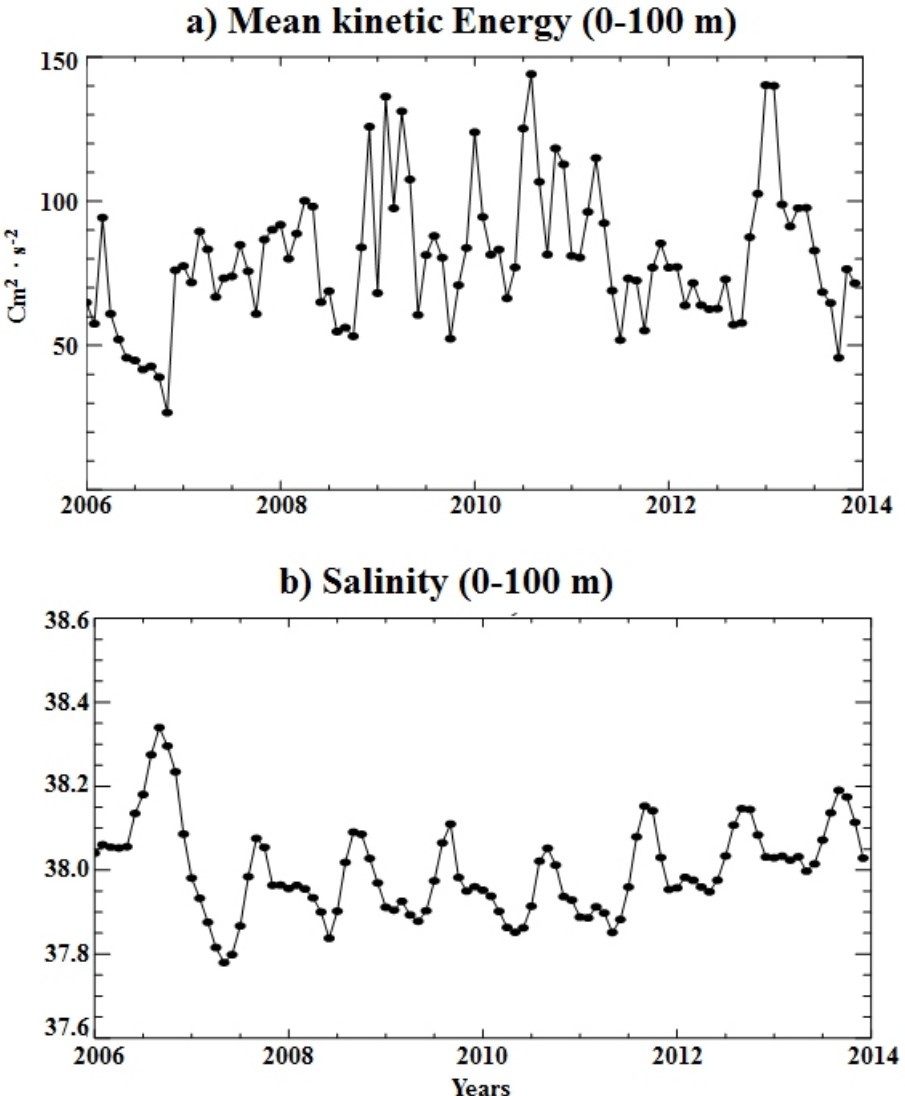

**Figure 6.** Monthly values of the Mean Kinetic Energy (MKE) (a) and of the salinity (b) (TYREM model), averaged over the southeastern TYS over the period of the cruises



**Table 1.** DOC average and stock (± uncertainties), calculated by vertical integration in the upper 50 m of each station, during the 8 cruises. For each cruise the average of the stocks, calculated taking into consideration all the sampled stations, is also reported, together with the standard deviation, the standard error and the total number of samples collected in each cruise. n.a.: data were not collected in these stations due to the bad weather conditions. For the position of the stations refer to the map reported in Figure 2.

| Station | Jan-09 | | Feb-10 | | Apr-07 | | Jun-07 | | Aug-10 | | Oct-10 | | Nov-06 | | Nov-11 | |
|---|---|---|---|---|---|---|---|---|---|---|---|---|---|---|---|---|
| | μM | mol·m⁻² | μM | mol·m⁻² | μM | mol·m⁻² | μM | mol·m⁻² | μM | mol·m⁻² | μM | mol·m⁻² | μM | mol·m⁻² | μM | mol·m⁻² |
| VTM5 | 53.3 | 2.66 | 58.7 | 2.93 | 59.1 | 2.96 | 64.3 | 3.21 | 61.6 | 3.08 | 67.5 | 3.38 | 62.5 | 3.13 | 68 | 3.4 |
| | ±0.6 | ±0.03 | ±0.6 | ±0.03 | ±1.1 | ±0.06 | ±1.5 | ±0.08 | ±0.5 | ±0.02 | ±0.7 | ±0.03 | ±0.8 | ±0.04 | ±0.5 | ±0.02 |
| VTM4 | 56.2 | 2.81 | n.a. | n.a. | n.a | n.a. | 65.1 | 3.25 | 64.4 | 3.22 | 67.4 | 3.37 | 61 | 3.05 | 68.8 | 3.44 |
| | ±0.3 | ±0.01 | | | | | ±0.8 | ±0.04 | ±0.7 | ±0.03 | ±0.9 | ±0.05 | ±0.8 | ±0.04 | ±0.5 | ±0.03 |
| VTM3 | 59 | 2.95 | n.a. | n.a. | 70.9 | 3.55 | 57.2 | 2.86 | 62 | 3.1 | 64.2 | 3.21 | 59.3 | 2.96 | 68 | 3.4 |
| | ±0.5 | ±0.02 | | | ±0.9 | ±0.04 | ±0.8 | ±0.04 | ±0.5 | ±0.03 | ±0.9 | ±0.05 | ±0.8 | ±0.04 | ±0.5 | ±0.03 |
| VTM2 | 57 | 2.85 | n.a. | n.a. | 69.7 | 3.49 | 61.1 | 3.06 | 62.1 | 3.11 | 64.5 | 3.22 | 56.7 | 2.84 | 59.1 | 2.95 |
| | ±0.8 | ±0.04 | | | ±1.3 | ±0.06 | ±0.8 | ±0.04 | ±0.9 | ±0.04 | ±0.8 | ±0.04 | ±1.3 | ±0.07 | ±0.6 | ±0.03 |
| VTM1 | 58.4 | 2.92 | 58.6 | 2.93 | n.a. | n.a. | 61.7 | 3.08 | 69.4 | 3.47 | 65 | 3.25 | 57.6 | 2.88 | 68.1 | 3.4 |
| | ±0.4 | ±0.02 | ±0.7 | ±0.04 | | | ±0.8 | ±0.04 | ±0.8 | ±0.04 | ±0.6 | ±0.03 | ±0.8 | ±0.04 | ±0.6 | ±0.03 |
| VTM | 57.4 | 2.87 | 58.8 | 2.94 | 61 | 3.05 | 65.1 | 3.26 | 66.8 | 3.34 | 62.7 | 3.14 | 58.5 | 2.92 | 60.7 | 3.04 |
| | ±0.7 | ±0.04 | ±0.9 | ±0.04 | ±0.9 | ±0.04 | ±1.0 | ±0.05 | ±1.0 | ±0.05 | ±0.6 | ±0.03 | ±1.4 | ±0.07 | ±0.4 | ±0.02 |
| Average | 56.9 | 2.84 | 58.7 | 2.93 | 65.2 | 3.26 | 62.4 | 3.12 | 64.4 | 3.22 | 65.2 | 3.26 | 59.3 | 2.96 | 65.4 | 3.27 |
| | ±0.6 | ±0.03 | ±0.7 | ±0.04 | ±1.0 | ±0.05 | ±0.9 | ±0.05 | ±0.7 | ±0.04 | ±0.7 | ±0.04 | ±1.0 | ±0.05 | ±0.5 | ±0.03 |
| St.Dev. | 2 | 0.10 | 0.1 | 0.0 | 6.4 | 0.32 | 3.1 | 0.15 | 3.1 | 0.16 | 1.9 | 0.09 | 2.2 | 0.1 | 4.3 | 0.2 |
| St.Err. | 0.8 | 0.04 | 0.1 | 0.0 | 3.2 | 0.16 | 1.3 | 0.06 | 1.8 | 0.09 | 1.1 | 0.05 | 0.9 | 0.04 | 1.8 | 0.1 |
| Samples | 23 | | 12 | | 13 | | 18 | | 24 | | 24 | | 18 | | 16 | |



**Table 2.** DOC inventory in the Tyrrhenian Sea. The surface area assumed for the Tyrrhenian Sea is $2.22 \cdot 10^5 \ Km^2$ (Astraldi and Gasparini, 1994). The volume of the deepest layer is estimated taking into consideration an average depth of $1477 \ m$, in agreement with Astraldi and Gasparini (1994). The mean DOC concentration is calculated taking into consideration all the data collected during the 8 cruises.

| Depth layer $[m]$ | Volume $[10^5 \ Km^3]$ | DOC Mean $\pm$ std $[\mu M]$ | Sample $[n]$ | Stock $[10^{12} \ g]$ |
|---|---|---|---|---|
| 0-50 | 0.111 | 63 ± 6 | 148 | 8.4±1 |
| 50-100 | 0.111 | 55 ± 5 | 88 | 7.3±0.8 |
| 100-500 | 0.888 | 45 ± 3 | 168 | 47.9±4 |
| 500-bottom | 3.28 | 41 ± 3 | 236 | 161±15 |
| Tot DOC Stock $[10^{12} \ g]$ | | | | 204-246 |

**Table 3.** DOC average concentrations and standard deviations, calculated by vertical integration in the upper $200 \ m$, Atlantic Water (AW) and DOC fluxes calculated for different periods in the study area (TYS), based on the water transports estimated by the model and DOC data collected during the 8 cruises.

| Area | Month | DOC [M] | Sample n. | AW Flux [Sv] | DOC Flux ($10^{12} \ g \ C \ yr^{-1}$) |
|---|---|---|---|---|---|
| TYS AW inflow | January 2009 | 51.5±1.5 | 41 | 1.33 | 25.9±0.8 |
| | February 2010 | 56.2±1.3 | 22 | 1.78 | 37.9±0.9 |
| | April 2007 | 58.3±5.4 | 25 | 1.09 | 24.1±2.2 |
| | June 2007 | 55.8±1.9 | 35 | 1.17 | 24.6±0.8 |
| | August 2010 | 55.2±2.3 | 47 | 0.48 | 10.1±0.4 |
| | October 2010 | 54.9±1.9 | 47 | 0.88 | 18.3±0.6 |
| | November 2006 | 51.6±1.3 | 35 | 0.45 | 8.8±0.2 |
| | November 2011 | 56.3±3.4 | 36 | 0.89 | 19.0±1.1 |