# Peer review of "Surface transport of DOC acts as a trophic link among Mediterranean sub-basins"

_Biogeosciences, 2018_

## Referee Comment (RC1) · Anonymous Referee #1 · 31 Oct 2018

The authors use data from a 6-station hydrographic section off the SW coast of Italy in the Tyrrhenian Sea to state that surface advection of Atlantic sourced water plays a crucial role in shaping DOC distribution in the Sea. This makes sense, of course, and is pretty obvious.

What I would find more interesting, if the authors agree, is the following set of processes: 1. DOC-enriched Atlantic Water is transported into the Tyrrhenian Sea where, in the net, it continues to escape remineralization while in the surface layer (as evidenced by the absence of change in concentrations or stock during summer). 2. It is then mineralized in support of upper mesopelagic microbes once winter overturn occurs. In this model, the DOM supporting the mesopelagic microbes is imported from outside the Tyrrhenian Sea, a story that is a bit more novel and defensible than the

one presented in the paper. This story remains consistent with the Conclusions of the manuscript.

Mechanism 1 for explaining the distribution of DOC concentrations: The authors write that "the anticyclone may determine dynamical regions where DOC can accumulate" But DOC doesn't "accumulate" by physical means, except for modest concentration by evaporation. Instead, DOC "accumulates" by biological processes. So I agree that the circulation will dictate where the DOC is present (in terms of elevated concentrations), but I do not understand how fluid trajectory controls "accumulation". The authors similarly wrote that "DOC is a passive tracer on the temporal scale of months; as such, its concentration can be strongly affected by the geometry of the advecting velocity field." This does not seem correct to me; the geometry will control distributions (which we see via concentrations), but it will not control (through alteration) concentration directly. The authors then direct the reader to Fig 5, where we see that the higher concentrations of DOC during August are at the ends of the section, where the authors see strong negative values of lambda. They say "strongly negative values of lambda {are} where the areal concentration can increase". Again, I do not see where stretching or broadening the trajectories of the surface flow (as lambda indexes) will actually change concentrations of DOC. Narrowing the flow of a specific water will reduce the spatial extent of the associated DOC (just as a river's spatial extent varies between broad and narrow sections along its path), but I do not see it changing concentrations in that flow. Perhaps I do not adequately understand the writing in this section. If so, the authors need to improve the clarity.

Mechanism 1 for shaping the horizontal distribution of DOC is explained in a long paragraph, but Mechanism 2 is not further addressed at all.

As for Section 3.5 "DOC annual cycle" 5/30: "DOC likely due to biological production resulting from the phytoplankton blooms in the Algerian Basin." There isn't really a way to know if the DOC was produced there, or somewhere further up stream, such as in the North Atlantic itself. The authors should tell us if the DOC entering the Med Sea

from the Atlantic is lower/higher/equal that present in AW present in the TYR; if higher, then a source in the Med Sea is required.

Specific Comments Page/Line 1/7: "flux...into the basin." It looks like the flux of DOM is out of the Algerian Basin and into the Tyrrhenian Sea, not into the basin (unless basin refers to TYR). 1/18: I suspect that Copin-Montegut and Avril 1993 would like to be recognized for their work in the Med as well. 3/15: what makes the cyclonic winds in the TYS "global"? I suggest deleting the word. 4/18: The Algerian Current is mentioned for the first time here, but not mentioned in the description of the system's circulation. 5/12: "to the northwest and south-southeast". It looks like the section runs from the NE to the SW, not NW to SW, and that is where the green colors under the section are located. 6/2: The term "the basin" has been used a few times in the text, but I'm not sure if this refers to the Algerian Basin or the basin holding the TYR. The authors need to be clear and consistent on that. 6/10: what is the "global cyclonic cell"? Should "washing" be "flushing"? 6/19: why is there a ? in parentheses?

Figures Figure 1: The arrows in Figure 1 used to identify straits are hard to see since they are black, as are the underlying current vectors. Also, I suggest that "ADT" be spelled out in the caption; I found myself having to look it up in the text to remember what it meant. The values for lat and long should include 'degree' symbols so that the reader knows what the values refer to. The caption needs to indicate which months were averaged for the winter and summer conditions. Figure 3: I cannot make out the velocity vectors in 1a, so I don't know which what the vectors are pointing. Units are not given for the vectors. In 3d, I wonder how well observed salinity matches with the modeled salinity shown in the plot. Figure 4: too many words in the caption are capitalized. What is "multi satellite"?

---

## Referee Comment (RC2) · Anonymous Referee #2 · 29 Nov 2018

The paper by Santinelli et al. presents results of Dissolved Organic Carbon (DOC) measurements in the Tyrrhenian area of the western Mediterranean Sea (MS), together with data from models and satellite observations. The overall goal is to explore the impact of the surface circulation on carbon dynamics in the western MS. According to the abstract, the main result is the quantification of the annual DOC input by the advection of Atlantic water (AW) in the Tyrrhenian area (TYS) of 8.8-37.9 10ˆ12 g DOC yr-1. Although it could be true that the advection of AW may play a crucial role in shaping DOC distribution in the TYS, I found the main result highly speculative and the paper although short, not well structured. I therefore will not recommend his publication in BG. I am not convinced that a strategy with a single transect as the one proposed here (Fig. 1) will allow to answer the question of the DOC entrance in the TYS with AW. The

very simple scheme presented in figure 1 considers no flow in the TYS during the summer periods. In paragraph 3, Line 19, it is indicated:" . . .a global anticyclonic circulation settles in the area, preventing the AW from entering the basin (Fig. 1b)". Paragraph 3.3 line 30, it is indicated: "In summer, the AW does not flow into the TYS. . .". In the 2 cases, there were no references to this statement. Nevertheless, the circulation in the Silicy Channel is highly complex and not very well described as documented by several papers (references below) and because it will affect the main result, it appears as a necessary minimum condition (not respected yet) to discuss about this initial postulate and how it may affect the main result. As written before, the paper is not well constructed and can be greatly improved regarding only the form. The beginning of the abstract is not correct for a scientific paper in BG. Many aspects presented in the result section of the paper are not mentioned in the abstract. Are they necessary to answer the central question raised? The paper is not easy to follow. Results and discussion are presented together. It seems necessary to present the results in a separate section and to take more attention on the presentation. The results section begins by a description of the hydrological context which is not a result from the paper! The result section contains some sentences that should appear in the method section. As an example, Line 26: "Doc averages, calculated by vertical integration in the upper 50 m. . .". It is not at the good place and not well explained. Using vertical integration, it is inventories and therefore it should be expressed in mol m-2, and not average concentrations expressed in $\mu$M. This is confusing. The discussion should be in a separate section and should focus on some aspects, as for example on the consequence of the simplification used for the seasonal circulation in the Silicy Channel indicating that there was no input of AW in the TYS during the summer periods (Figure 1). Regarding the presentation, DOC measurements are presented using a color bar on Fig. 2 and with symbols and unexplained ranges on Fig. 3d (70-73 $\mu$M; 63-64 $\mu$M; 56-60 $\mu$M; 47-53 $\mu$M): why? It is necessary to be consistent in order to help the readers, and, as an example, adding isohalines on a DOC section could may be more helpful to present salinity gradients in the present case. Page 5 Line 18: "As previously reported, DOC

distribution is markedly different in November 2006 and 2011. Average DOC concentrations (0−50 m) are ≈ 6 $\mu$M lower in 2006 than in 2011, and this difference results in ≈ 0.31 mol Âům−2 reduction in the 0−50 m accumulated stocks (Table 1)". I don't understand. It is lower in 2006 than in 2011, and therefore, it is increasing!? In Table 1, the first column presents DOC concentration from Jan 2009, the third column from Apr 2007... I finally understand that the month was more important than the year for the authors... but it is clearly not straightforward as many other points in the ms. Regarding the large interannual variability, are you sure you will be able to show seasonal variations?... and add a DOC annual cycle section (3.5). January 2009 is clearly different (Fig. 2) but are the others DOC sections presented different? At the end of the ms, the authors propose DOC budget at the scale of the TYS with taking into consideration all the data collected during the 8 cruises on the same section (Fig. 1). It is although highly speculative and need at least to be discussed. DOC measurements are scarce in the MS and it will be of interest to publish these data. Nevertheless, I will encourage the author to find a better way to do it.

References: Rio, M.-H., Pascual, A., Poulain, P.-M., Menna, M., Barceló, B., and Tintoré, J.: Computation of a new mean dynamic topography for the Mediterranean Sea from model outputs, altimeter measurements and oceanographic in situ data, Ocean Sci., 10, 731-744, https://doi.org/10.5194/os-10-731-2014, 2014. Jouini, M., K. Béranger, T. Arsouze, J. Beuvier, S. Thiria, M. Crépon, and I. Taupier-Letage (2016), The Sicily Channel surface circulation revisited using a neural clustering analysis of a high-resolution simulation, J. Geophys. Res. Oceans, 121, doi:10.1002/2015JC011472. Gerin R., Poulain P.-M., Taupier-Letage I., Millot C., Ben Ismail S. and C. Sammari, 2009. Surface circulation in the Eastern Mediterranean using Lagrangian drifters (2005-2007). Ocean Science, 5, 559–574.

———————————————

---

## Author Comment (AC1) · 22 Dec 2018

The authors use data from a 6-station hydrographic section off the SW coast of Italy in the Tyrrhenian Sea to state that surface advection of Atlantic sourced water plays a crucial role in shaping DOC distribution in the Sea. This makes sense, of course, and is pretty obvious.

Even if we agree with the referee that it is quite obvious that dissolved organic carbon (DOC) is transported, as many other tracers, by water, we believe that: (1) the importance of this process is overlooked in the literature and in the models, which generally focus on local primary production and (2) compared to internal production, the relative weight of external DOC input is noteworthy and needs to be correctly quantified, es-

pecially because it certainly occurs at larger scale in the open ocean (see below). It is well known that lateral advection affects the spatial distribution of dissolved substances in the ocean; however, very few works have been devoted to study and quantify this process for DOC which, in oligotrophic environments, account for the largest fraction of utilizable reduced carbon (e.g., Santana-Falcón et al., 2016; Wu et al., 2015; Hansell et al. 1997). Our study addresses this process for the first time in the Mediterranean Sea and shows that horizontal transport of DOC into the Tyrrhenian Sea is of the same order of magnitude, or even larger, than the in-situ DOC production. The small size of the Mediterranean Sea allows for short transfer times which, in turn, favors the preservation of the DOC stock produced elsewhere. However, this transfer may be important also in other regions of the oceans. In this scenario surface advection may set up, via horizontal transfer, a sort of compensation among regions with different trophic regimes, smoothing trophic gradients. We believe that understanding these transport processes is a crucial and preliminary step to understand and quantify all the other processes (biological, chemical, geological) that influence DOC distribution on a variety of timescales.

What I would find more interesting, if the authors agree, is the following set of processes: 1. DOC-enriched Atlantic Water is transported into the Tyrrhenian Sea where, in the net, it continues to escape remineralization while in the surface layer (as evidenced by the absence of change in concentrations or stock during summer).

We suspect that DOC remineralization is reduced in summer, as also proposed by Santinelli et al., (PiO, 119, 68-77, 2013), but, as we clarify in the text: "Even if DOC concentration does not change, DOC turnover can occur either by (1) advected DOC is consumed and replaced by DOC released within the basin or coming from external sources or (2) removal of DOC released within the basin or coming from external sources and aging of the advected DOC."

2. It is then mineralized in support of upper mesopelagic microbes once winter overturn occurs. In this model, the DOM supporting the mesopelagic microbes is imported from

outside the Tyrrhenian Sea, a story that is a bit more novel and defensible than the one presented in the paper. This story remains consistent with the Conclusions of the manuscript.

We thank the referee for this suggestion. The DOC redistributed by mixing in the fall-winter period feeds the mesopelagic microbes and we find an interesting hypothesis that the mesopelagic communities can use the DOC coming from outside the Tyrrhenian Sea instead of the surface communities. In addition, fall-winter is the time when the Tyrrhenian circulation 're-opens' thus restoring a significant exchange with the Ligurian sea; a fraction of the surface DOC can therefore be exported northward contributing to the large amount of DOC exported to depth by deep convection, feeding the deep water microbes in the Western Mediterranean Sea (see Santinelli et al., 2010; Christensen, J. P.et al. GBCycles, 3(4), 315-335, 1989). On the other hand we do not agree with the fact the showing that the DOC supporting the mesopelagic microbes is imported from outside the Tyrrhenian Sea, would be a story a bit more novel and defensible that the fact that advection represents an important and overlooked source of DOC to the Tyrrhenian Sea and that it can fuel the microbial loop in the surface layer. We can rework the discussion including this hypothesis.

Mechanism 1 for explaining the distribution of DOC concentrations: The authors write that "the anticyclone may determine dynamical regions where DOC can accumulate" But DOC doesn't "accumulate" by physical means, except for modest concentration by evaporation. Instead, DOC "accumulates" by biological processes. So I agree that the circulation will dictate where the DOC is present (in terms of elevated concentrations), but I do not understand how fluid trajectory controls "accumulation". The authors similarly wrote that "DOC is a passive tracer on the temporal scale of months; as such, its concentration can be strongly affected by the geometry of the advecting velocity field." This does not seem correct to me; the geometry will control distributions (which we see via concentrations), but it will not control (through alteration) concentration directly. The authors then direct the reader to Fig 5, where we see that the higher concentrations of DOC during August are at the ends of the section, where the authors see strong negative values of lambda. They say "strongly negative values of lambda {are} where the areal concentration can increase". Again, I do not see where stretching or broadening the trajectories of the surface flow (as lambda indexes) will actually change concentrations of DOC. Narrowing the flow of a specific water will reduce the spatial extent of the associated DOC (just as a river's spatial extent varies between broad and narrow sections along its path), but I do not see it changing concentrations in that flow. Perhaps I do not adequately understand the writing in this section. If so, the authors need to improve the clarity.

The referee is absolutely right for what concerns the DOC concentration in each small water parcel. By local accumulation here we refer to the concentration of DOC, over a spatial scale encompassing the velocity field at meso/large-scale. On such scales, DOC, considered as a passive tracer, can be redistributed depending on the geometry of the advecting velocity field, and can thus locally increase in a specific area, even if the water concentration does not increase. The problem of quantifying local changes of passive tracer concentration in a stationary or weakly changing flow is a classical problem of geophysical fluid dynamics, which has been revisited in Haller-Iacono (2003), where new quantitative measures of these effects (e.g., the "lambda parameter") have been introduced. In other words, we states that the DOC spatial maxima observed do not result only by local production but by the geometry of the flow. We will better clarify this concept in the revised version.

Mechanism 1 for shaping the horizontal distribution of DOC is explained in a long paragraph, but Mechanism 2 is not further addressed at all.

We explained mechanism 1 more in depth since it is more difficult to be understood and introduces a new idea. We think that Mechanism 2 is pretty easy to understand and without further data it cannot be tested.

As for Section 3.5 "DOC annual cycle" 5/30: "DOC likely due to biological production

resulting from the phytoplankton blooms in the Algerian Basin." There isn't really a way to know if the DOC was produced there, or somewhere further up stream, such as in the North Atlantic itself. The authors should tell us if the DOC entering the Med Sea from the Atlantic is lower/higher/equal that present in AW present in the TYR; if higher, then a source in the Med Sea is required.

DOC concentration in the AW close to the Gibraltar Strait is highly variable, but lower than in the Tyrrhenian Sea: 60 ± 4 $\mu$M in April 1998 (Dafner et al., 2001), 51-54 $\mu$M in September 1999 (Santinelli et al., 2013), and 50-60 $\mu$M in May/June 2007. So yes, DOC in the AW core is lower when it enters the Med Sea than in the Tyr Sea, suggesting that it is enriched in DOC during its route. We can add this information in the revised manuscript.

Specific Comments Page/Line 1/7: "flux. . .into the basin." It looks like the flux of DOM is out of the Algerian Basin and into the Tyrrhenian Sea, not into the basin (unless basin refers to TYR).

Yes, the basin refers to the TYS (this is the acronym we have used for the Tyrrhenian Sea), we will clarify.

1/18: I suspect that Copin-Montegut and Avril 1993 would like to be recognized for their work in the Med as well.

We will add the reference

3/15: what makes the cyclonic winds in the TYS "global"? I suggest deleting the word.

We have used the word "global" to indicate that the large-scale wind stress is cyclonic over most of the TYS; this can be seen, for example, in Figure 2 of Iacono et al. (2013). This global wind stress drives the cyclonic circulation along the Italian coast, from Sicily to the Corsica Channel, which is one of the distinctive features of the winter-spring dynamics. If less confusing we can replace "global" by "basin-scale"

4/18: The Algerian Current is mentioned for the first time here, but not mentioned in

the description of the system's circulation.

The description of the main pattern of circulation, including the Algerian Current, will be added at the end of the introduction or in a specific section.

5/12: "to the northwest and south-southeast". It looks like the section runs from the NE to the SW, not NW to SW, and that is where the green colors under the section are located.

The referee is right, we apologize for the mistake.

6/2: The term "the basin" has been used a few times in the text, but I'm not sure if this refers to the Algerian Basin or the basin holding the TYR. The authors need to be clear and consistent on that.

OK, we will clarify.

6/10: what is the "global cyclonic cell"? Should "washing" be "flushing"?

As for the "global cyclonic cell", see answer to 3/15. Yes, flushing is more appropriate. We will correct the sentence.

6/19: why is there a ? in parentheses?

This is a typo. The ? will be removed.

Figures Figure 1: The arrows in Figure 1 used to identify straits are hard to see since they are black, as are the underlying current vectors.

Ok

Also, I suggest that "ADT" be spelled out in the caption; I found myself having to look it up in the text to remember what it meant.

Ok

The values for lat and long should include 'degree' symbols so that the reader knows

what the values refer to.

Ok

The caption needs to indicate which months were averaged for the winter and summer conditions.

Winter refers to January-March, and summer to July-September. We will clarify this in the caption.

Figure 3: I cannot make out the velocity vectors in 1a, so I don't know which what the vectors are pointing.

Ok, we will make a new figure with a bigger panel devoted to the circulation.

Units are not given for the vectors.

There is a reference arrow in the upper part of the figure that corresponds to 0.5 m/s.

In 3d, I wonder how well observed salinity matches with the modeled salinity shown in the plot.

Comparing the salinity section (model) and the T-S diagram, where the depths of the observations are marked by red numbers, one can see that there is good correspondence between the model and measured values.

Figure 4: too many words in the caption are capitalized.

Ok

What is "multi satellite"?

Multi-satellite means that the daily maps are obtained merging measurements made by different satellites.

Please also note the supplement to this comment:
https://www.biogeosciences-discuss.net/bg-2018-418/bg-2018-418-AC1-

supplement.pdf

---

## Author Comment (AC2) · 22 Dec 2018

The paper by Santinelli et al. presents results of Dissolved Organic Carbon (DOC) measurements in the Tyrrhenian area of the western Mediterranean Sea (MS), together with data from models and satellite observations. The overall goal is to explore the impact of the surface circulation on carbon dynamics in the western MS. According to the abstract, the main result is the quantification of the annual DOC input by the advection of Atlantic water (AW) in the Tyrrhenian area (TYS) of 8.8-37.9 $10^{12}$ g DOC yr-1.

We will rephrase the abstract to better clarify that the main result of the paper is that lateral advection of the AW plays a crucial role in regulating DOC concentration and distribution in the Tyrrhenian Sea and that horizontal transport of DOC into the Tyrrhenian Sea is of the same order of magnitude, or even larger, than the in-situ DOC production. Our study addresses this process for the first time in the Mediterranean Sea. The small size of the basin allows for short transfer times which, in turn, favors the preservation of the DOC stock produced elsewhere. However, this transfer may be important also in other regions of the oceans. In this scenario surface advection may set up, via horizontal transfer, a sort of compensation among regions with different trophic regimes, smoothing trophic gradients. We believe that understanding these transport processes is a crucial and preliminary step to understand and quantify all the other processes (biological, chemical, geological) that influence DOC distribution on a variety of timescales.

Although it could be true that the advection of AW may play a crucial role in shaping DOC distribution in the TYS, I found the main result highly speculative and the paper although short, not well structured. I therefore will not recommend his publication in BG.

We definitely disagree with the fact that the result is highly speculative. Coupling circulation patterns and transport, derived by a fully validated, basin-scale hydrodynamic model, and repeated DOC data on a 6-station section, we believe that we have provide convincing evidence that AW advection plays a crucial role in shaping DOC stocks and distribution in the basin. Of course, there are uncertainties in the total amounts, which we have discussed in the text but the fact the highest DOC transport is associated to the AW is clearly showed by the data. In addition, this section of the Mediterranean Sea is a very dynamic, but tractable, study site for understanding biogeochemical process that could, in principle, operate over broader scales of time and space. We do not understand either why the paper is not well structured. We agree that some paragraphs might be moved in other sections and that other ones can be improved in clarity, but it is difficult to understand why the paper, after better clarifying the parts that have raised perplexities in the referees, could not be published.

I am not convinced that a strategy with a single transect as the one proposed here (Fig. 1) will allow to answer the question of the DOC entrance in the TYS with AW.

We obviously agree with the referee that a larger data set would reinforce our conclusions, but we disagree that a single transect is not the proper sampling scheme when the issues at stake are water and tracers transport. Indeed, Geosecs, WOCE and, more recently, Geotraces have all organized their sampling along 'single' transects. The referee should also acknowledge that our DOC data-set has a good spatial and temporal resolution, compared to what is available in the literature. A DOC data-set with a 6-station section repeated for 8 times in different seasons and years is unique and precious, nothing similar is available for the Med Sea nor for the Oceans. There are fixed stations, that cannot give a 3D idea of DOC distribution. In addition the position of the section is crucial, since it located in an area of the Tyrrhenian Sea where a strong stream of AW of remote origin is present every year, for half of the year. We therefore strongly believe that our data clearly shows the link between DOC surface distribution and AW advection and that this work could be the first one to address this overlooked mechanisms opening to further investigation in the field.

The very simple scheme presented in figure 1 considers no flow in the TYS during the summer periods. In paragraph 3, Line 19, it is indicated:" . . .a global anticyclonic circulation settles in the area, preventing the AW from entering the basin (Fig. 1b)". Paragraph 3.3 line 30, it is indicated: "In summer, the AW does not flow into the TYS. . .". In the 2 cases, there were no references to this statement.

Figure 1 does not present a scheme of the circulation; the white arrows are only added to further highlight the different paths of the AW in winter and summer. We should have better explained the content of this figure in the text. The figure represents a scientific result, since it displays the winter and summer maps of the Absolute Dynamic Topography (ADT; indicated by colors) that we have obtained by averaging over 21 years of AVISO altimeter data (1993-2013), with the corresponding average geostrophic circulations (also derived from the satellite data) superposed. The maps clearly show a

robust AW stream circling cyclonically all around the Italian coasts in winter (actually from the end of the autumn to mid spring), which is absent in summer. Looking at individual years, one always gets the same picture. This seasonal difference in the surface circulation of the TYS is known since the seminal works by the Russians in the eighties (e.g., Krivosheya, 1983), and has been further analyzed in more recent investigations (Rinaldi et al., 2010; Iacono et al., 2013; Napolitano et al., 2014)

Nevertheless, the circulation in the Sicily Channel is highly complex and not very well described as documented by several papers (references below) and because it will affect the main result, it appears as a necessary minimum condition (not respected yet) to discuss about this initial postulate and how it may affect the main result.

It is not clear to us why the referee is so concerned about the complexity of the circulation in the Sicily Channel, and which postulate he/she is referring to. The complexity and variability of the circulation in the Sicily Channel does not alter the well-known fact that the AW does not enter the TYS in summer (see previous point). We note that, in any case, the main point of the paper is about what happens in winter and beginning of spring, months in which a robust stream of AW (of the strength of about 1 Sv, according to both observations and numerical results) enters the TYS.

As written before, the paper is not well constructed and can be greatly improved regarding only the form. The beginning of the abstract is not correct for a scientific paper in BG. Many aspects presented in the result section of the paper are not mentioned in the abstract. Are they necessary to answer the central question raised?

We are quite surprised by the statement "Characterizing carbon cycling and redistribution in the ocean is an important issue for Mankind, because it may affect key ecosystem services, e.g., support to climate system and food provision." is not correct for a scientific paper in BG. Carbon cycle is likely the most discussed theme in Earth System Science in the last decades and the main reason is that its dynamics impact, directly or indirectly, several aspects of human life. The statement was made to frame our study

in a highly debated issue. Along with the comments of Ref#1 we will rephrase and improve the abstract.

The paper is not easy to follow. Results and discussion are presented together. It seems necessary to present the results in a separate section and to take more attention on the presentation.

We do not understand what the reviewer means with "to take more attention on the presentation of the results", we have the feeling that he/she read the paper very quickly, since all the main results are carefully descripted and presented. We do not understand either why the paper is not well constructed. We decided to keep together results and discussion, because the results are briefly presented in section 3.1, whereas in 3.2, 3.3 and 3.4 we try to combine in-situ data with model results in order to facilitate the discussion of the results. In our opinion, it is therefore helpful for the reader to have data and discussion combined in the same sections. However, if the editor agrees with this comment, it is not a problem to separate results and discussion.

The results section begins by a description of the hydrological context which is not a result from the paper!

As reported in the text, that section was added to recall some basic elements of the surface circulation in the WMED. However, in agreement with this comment we can move this section to the introduction or add a section dedicated to explain in details the main features of the TYS circulation, before the results.

The result section contains some sentences that should appear in the method section. As an example, Line 26: "Doc averages, calculated by vertical integration in the upper 50 m.". It is not at the good place and not well explained. Using vertical integration, it is inventories and therefore it should be expressed in mol m-2, and not average concentrations expressed in $\mu$M. This is confusing.

We agree with this comment, the sentence is not clear. The average concentration was

calculated dividing the inventories, calculated by vertical integration in the 0-50 m, by the depth (50 m) of the layer. We can easily move this sentence in the material and method section explaining better the calculation.

The discussion should be in a separate section and should focus on some aspects, as for example on the consequence of the simplification used for the seasonal circulation in the Silicy Channel indicating that there was no input of AW in the TYS during the summer periods (Figure 1).

We do not understand why the referee keeps insisting on the role of the Sicily Channel. The fact that the AW cannot penetrate into the TYS in summer is due to the local dynamics of the TYS, since in this season a broad anticyclonic circulation forms in the southern part of the basin (see, e.g, Krivosheya, 1983; Artale et al., 1994; Marullo et al., 1994; Pierini and Simioli, 1998; Korres et al., 2000; Iacono et al., 2013; Napolitano et al., 2014; among others). In any case, what is more important is what happens in winter and spring, when the AW inflow takes place, so we do not see why we should further focus on the summer dynamics and on what happens in the Sicily Channel.

Regarding the presentation, DOC measurements are presented using a color bar on Fig. 2 and with symbols and unexplained ranges on Fig. 3d (70-73 $\mu$M; 63-64 $\mu$M; 56-60 $\mu$M; 47-53 $\mu$M): why? It is necessary to be consistent in order to help the readers, and, as an example, adding isohalines on a DOC section could may be more helpful to present salinity gradients in the present case.

We totally disagree with this comment. We think that data presentation is consistent, we are just trying to help the readers giving a different way to look at the data. In fig 3d we try to give an integrated view of salinity distribution, current velocity and DOC distribution, in order to show how DOC distribution is shaped by AW flow. Even if we agree that the figure needs a little attention to be understood, it is not so complicated and after a lot of work this was the best way we found to give a complete and integrated view of DOC, salinity and velocity along this section. We are happy to try other

way to represent these data if the reviewer gives us a constructive suggestion. In the DOC vertical distribution we added isohalines because they are directly related to the circulation.

Page 5 Line 18: "As previously reported, DOC distribution is markedly different in November 2006 and 2011. Average DOC concentrations (0−50 m) are ≈ 6 $\mu$M lower in 2006 than in 2011, and this difference results in ≈ 0.31 mol AÌĆu ÌŁm−2 reduction in the 0−50 m accumulated stocks (Table 1)". I don't understand. It is lower in 2006 than in 2011, and therefore, it is increasing!?

We do not understand this comment. We say that DOC concentrations (0−50 m) are ≈ 6 $\mu$M and the stocks are 0.31 mol m−2 lower in 2006 than in 2011, we did not say that the DOC concentration is increasing.

In Table 1, the first column presents DOC concentration from Jan 2009, the third column from Apr 2007. . .. I finally understand that the month was more important than the year for the authors. . .. but it is clearly not straightforward as many other points in the ms.

We gave more importance to the months because the Tyrrhenian sea displays a bimodal physical dynamics that depends on seasons. Therefore is the seasonal signal that must be captured at the best, within the limits of our data-set. The interannual variability, which we also could relate to the interannual variation in the physical dynamics (see below), was not the scope of this study.

Regarding the large interannual variability, are you sure you will be able to show seasonal variations?... and add a DOC annual cycle section (3.5). January 2009 is clearly different (Fig. 2) but are the others DOC sections presented different?

Indeed, the difference between DOC distribution and concentration in November 2006 and 2011 is clearly due to the anomalous pattern of circulation observed in 2006. In our opinion the annual cycle of DOC is clearly visible in figure 2 and Table 1, as clarified

in the text

At the end of the ms, the authors propose DOC budget at the scale of the TYS with taking into consideration all the data collected during the 8 cruises on the same section (Fig. 1). It is although highly speculative and need at least to be discussed.

We can rework the section discussing the limitation of our estimate. Although speculative we believe that these calculations provide substantial evidence of the relevance of AW input in DOC dynamics in the TYS.

DOC measurements are scarce in the MS and it will be of interest to publish these data. Nevertheless, I will encourage the author to find a better way to do it.

We thank the referee for the acknowledgement and we believe that our analysis is robust and gives convincing evidence that AW advection plays a crucial role in shaping DOC stocks and distribution in the basin. However, we will be happy to follow any constructive suggestion that could further improve the quality of this study.

Please also note the supplement to this comment:
https://www.biogeosciences-discuss.net/bg-2018-418/bg-2018-418-AC2-supplement.pdf